# CROSSCODEEVAL: A Diverse and Multilingual Benchmark for Cross-File Code Completion

**Yangruibo Ding**[1]* **Zijian Wang**[2,*] **Wasi Uddin Ahmad**[2,*]
**Hantian Ding**[2] **Ming Tan**[2] **Nihal Jain**[2] **Murali Krishna Ramanathan**[2]
**Ramesh Nallapati**[2] **Parminder Bhatia**[2] **Dan Roth**[2] **Bing Xiang**[2]

[1]Columbia University [2]AWS AI Labs

yrbding@cs.columbia.edu  {zijwan,wuahmad}@amazon.com

https://crosscodeeval.github.io

## Abstract

Code completion models have made significant progress in recent years, yet current popular evaluation datasets, such as HumanEval and MBPP, predominantly focus on code completion tasks within a single file. This over-simplified setting falls short of representing the real-world software development scenario where repositories span multiple files with numerous cross-file dependencies, and accessing and understanding cross-file context is often required to complete the code correctly.

To fill in this gap, we propose CROSSCODEEVAL, a diverse and multilingual code completion benchmark that necessitates an in-depth cross-file contextual understanding to complete the code accurately. CROSSCODEEVAL is built on a diverse set of real-world, open-sourced, permissively-licensed repositories in four popular programming languages: Python, Java, TypeScript, and C#. To create examples that strictly require cross-file context for accurate completion, we propose a straightforward yet efficient static-analysis-based approach to pinpoint the use of cross-file context within the current file.

Extensive experiments on state-of-the-art code language models like CodeGen and StarCoder demonstrate that CROSSCODEEVAL is extremely challenging when the relevant cross-file context is absent, and we see clear improvements when adding these context into the prompt. However, despite such improvements, the pinnacle of performance remains notably unattained even with the highest-performing model, indicating that CROSSCODEEVAL is also capable of assessing model's capability in leveraging extensive context to make better code completion. Finally, we benchmarked various methods in retrieving cross-file context, and show that CROSSCODEEVAL can also be used to measure the capability of code retrievers.

## 1 Introduction

Language models for code (code LMs), such as Codex (Chen et al., 2021), CodeGen (Nijkamp et al., 2023b,a), and StarCoder (Li et al., 2023), have demonstrated their power to enhance developer productivity through their promising results in code completion tasks. To evaluate these models, researchers propose multiple code completion evaluation benchmarks, (e.g., Chen et al., 2021; Lu et al., 2021; Athiwaratkun et al., 2023; Austin et al., 2021), where the model is asked to complete the code given the context in the current file. However, such an evaluation setting is over-simplified, and it is not able to reflect the model's capability in code completion accurately. Specifically, in the realm of modern software development, repositories consist of multiple files, each interwoven with

---

*  Equal Contribution. Work done while Yangruibo Ding was an intern at AWS AI Labs.

37th Conference on Neural Information Processing Systems (NeurIPS 2023) Track on Datasets and Benchmarks.

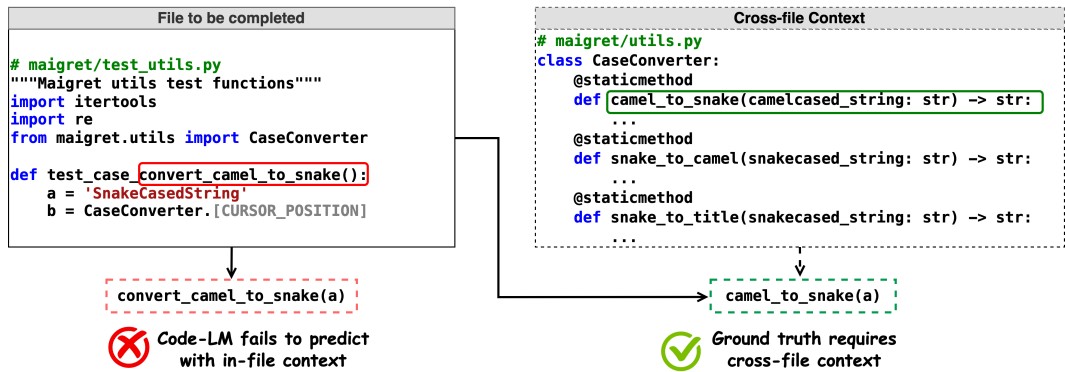

Figure 1: Code LM fails to complete a Python test case since the in-file context (left figure) does not provide sufficient information. The function name from the current file indicates that the completing function is a test case for `convert_camel_to_snake`, so with only such context, the model hallucinates wrong completion as `convert_camel_to_snake`. However, the failure is not due to the model's capacity, but the necessary cross-file context is not present (right figure). When the class `CaseConverter` is present in the prompt, the model generates `camel_to_snake` correctly.

extensive cross-file dependencies, i.e., contextual information from other source code files within the same repository. A significant drawback of most existing benchmarks is their tendency to overlook these complex dependencies. As a result, they fall short of providing a comprehensive evaluation of code completion models within realistic real-world scenarios.

Figure 1 illustrates the limitation of common code completion evaluation sets with a real example. The developer is writing a test case for a class, `CaseConverter`, implemented within the current repository. CodeGen-2B-mono (Nijkamp et al., 2023b), a large Python code LM, fails to complete the API call if only the current file context is present.

Motivated by such examples and to fill in the need of evaluating code completion in realistic software development with numerous intervening cross-file context dependencies, we propose CROSSCODEE-VAL, a diverse and multilingual benchmark to evaluate code language models' ability to use cross-file context for code completion. This new dataset is composed of 10k examples from 1k repositories in 4 languages. Unlike existing datasets where the correct answer could be predicted with only context from the current file, CROSSCODEEVAL strictly requires cross-file context to correctly complete the missing code (§2.2). CROSSCODEEVAL's examples are carefully curated from existing open-sourced repositories with a series of quality filters, and we ensure CROSSCODEEVAL has minimal overlap with the training data from existing code LMs, eliminating the confounder of data leakage and memorization in result interpretation (§2.1 & 2.3).

We conducted a comprehensive evaluation of popular public and proprietary code LMs: CodeGen (Nijkamp et al., 2023b,a) and StarCoder (Li et al., 2023) in various sizes from 350M to 16B parameters, and OpenAI's GPT-3.5-Turbo in §3. Empirical results reveal that when given *only* the current-file context, these models yield suboptimal results. Remarkably, incorporating cross-file context into the prompt significantly enhances the performance of these code LMs, even in a zero-shot setting. This underscores that CROSSCODEEVAL effectively serves its goal as a benchmark aimed at evaluating cross-file code completion. Moreover, even when providing cross-file context in the prompt, the performance of the most powerful models remains notably imperfect, highlighting that CROSSCODEEVAL is also instrumental in assessing a model's ability for leveraging extensive context in code completion. Lastly, we benchmarked various retrieval methods from sparse to dense, demonstrating that CROSSCODEEVAL can additionally serve as a benchmark for code retrieval.

## 2 CROSSCODEEVAL: A Benchmark for Cross-File Code Completion

CROSSCODEEVAL is a diverse and multilingual scope completion dataset in four popular languages: Python, Java, TypeScript, and C# where examples include code prompts ending in an imagined cursor position and references include the code token sequences from the cursor position to the end of the statement. CROSSCODEEVAL examples have a key property - the statement to be completed must

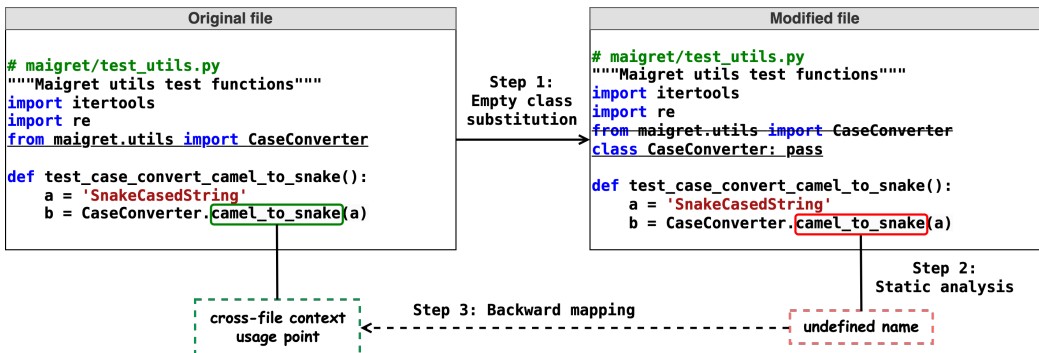

Figure 2: We replace the third import statement, which is from the same repository, with an empty class. Consequently, `camel_to_snake` in the last line becomes an undefined name in the modified file. Thereby, we know this method in the original file is defined only in the cross-file context.

have at least one use of local API (classes, variables, and methods defined in the software repository). Next, we briefly describe how we collect the software repositories (§2.1), select a subset of them for CROSSCODEEVAL construction (§2.2), post-processing and quality control process (§2.3), and CROSSCODEEVAL statistics and future scope for extending the dataset (§2.4).

## 2.1 Dataset Collection

We collect permissively licensed repositories from GitHub. To mitigate potential data leakage issues,[2] we focus on repos that were created *recently* and not forks. Specifically, we collected repos created between 2023-03-05 to 2023-06-15 on 2023-09-01. The time span ensures sufficient data collected with no overlap with the training data of many existing code LMs released before mid-2023, no matter whether the data is publicly available or not. We limit repos to contain the four languages we study and we keep only repos with zipped file size < 1MB and number of stars >= 3. Then we filter out repos that have fewer than 10 or more than 50 source code files. Finally, we remove the repos with at least one source code file that exactly matches one of the code files in the commonly used Stack (Kocetkov et al., 2022) dataset. As a result, we ended up with 471, 239, 193, and 99 repos, respectively.

## 2.2 Dataset Generation

We propose a static-analysis-based method to identify code fragments that require cross-file context automatically. Our approach is illustrated in Fig 2. First, we find all intra-project imports in the original file. Next, an empty class is created for each imported name to replace the import statement. Since the imported name now refers to an empty class, any subsequent call to its member function or attribute will raise an undefined name error. We leverage static analysis to catch such errors in the modified file, which precisely correspond to the names in the original file that can only be resolved by cross-file context. We map the location of undefined names back to the original file to determine the split point of the prompt and the reference.

To increase the variety in the dataset, we randomly select a tree-sitter[3] token in the same line before the cross-file entity to be the cursor location, splitting the code to a prompt and a reference. It is often the case that the same cross-file API is called multiple times in a file, and there is a chance for models to infer the API name from previous calls even without cross-file context. Therefore, if the same undefined name is reported at multiple places in a file, we keep only the first occurrence. In this work, we focus on instantiating our approach in the four popular programming languages, while the idea can be generalized to other languages in principle. Specifically, for Python, we use `Pylint`[4] to detect undefined names; for Java, we use `javac` compiler; for TypeScript, we use `tsc` compiler;

---

[2]Overlapping between CROSSCODEEVAL and data used to pretrain code LMs.

[3]https://tree-sitter.github.io/tree-sitter/

[4]Pylint is a static code analyzer for Python: https://pylint.readthedocs.io/en/latest/

for C#, we use `csc` compiler from the `mono`[5] image. We use tree-sitter to identify full statements to construct reference completions in Python. For Java, TypeScript, and C#, we consider statements ending with either ";", "{", or "}". See Appendix A for more details.

## 2.3 Post-processing and Quality Control

We designed a series of rule-based and model-based post-processing filters to ensure the quality of the dataset. We filter examples if (1) fewer than $N$ lines of code (lines not including import statements, where $N = 10, 20, 30, 5$ for Python, Java, TypeScript, and C#) in the prompt, (2) too short ($< 3$ tokens), or long ($> 30$ tokens) reference. We exclude examples if the references are found verbatim in any other source code file within the repository (*i.e.,* cross-file). We further discard examples with duplicate references. The filtering steps collectively remove 15%-20% of the examples.

Moreover, to ensure that the reference isn't predictably inferred solely from the current file (possibly owing to strong clues in function names and comments), we feed the examples (input prompts) to `starcoderbase-1B` model (Li et al., 2023) to complete the statement and remove the exact matches. This step results in the removal of $<10\%$ of the generated examples. As an ancillary benefit, this further safeguards that the examples are not seen by publicly available code LMs while CROSSCODEEVAL is constructed based on repositories that do not overlap with the Stack and possibly other private pre-training datasets created prior to 2023. Finally, we perform human annotations on a subsample of the resulting CROSSCODEEVAL and found that the dataset has a satisfactory quality to serve the goal of cross-file code completion. See Appendix B for more details.

## 2.4 Dataset Statistics, Scope, and Future Extensions

**Statistics** We present the statistics of CROSSCODEEVAL in Table 1. We use the StarCoder tokenizer (Li et al., 2023) to compute the number of tokens.

| Feature | Python | Java | TypeScript | C# |
|---|---|---|---|---|
| # Repositories | 471 | 239 | 193 | 99 |
| # Files | 1368 | 745 | 779 | 642 |
| # Examples | 2665 | 2139 | 3356 | 1768 |
| Avg. # lines in prompt | 90.6 | 106.7 | 116.5 | 71.1 |
| Avg. # tokens in prompt | 938.9 | 995.3 | 944.9 | 584.1 |
| Avg. # lines in reference | 1.0 | 1.1 | 1.7 | 1.7 |
| Avg. # tokens in reference | 13.2 | 14.5 | 17.4 | 12.5 |

Table 1: CROSSCODEEVAL statistics.

**Scope** In addition to prompts and references, we include the code lines that follow the references from the original source code files in CROSSCODEEVAL examples. Given the source code lines to the left (prompt or prefix) and right (suffix) of the references, CROSSCODEEVAL can be used to evaluate code LMs for their fill-in-the-middle (FIM) capabilities (Bavarian et al., 2022).

**Future Extensions** CROSSCODEEVAL currently supports four popular languages. As our method is generalizable, CROSSCODEEVAL can potentially be extended to other languages. Additionally, we advise future code LM pre-training datasets should explicitly exclude CROSSCODEEVAL to minimize the effect of memorization.

# 3 Experiments

## 3.1 Models

We benchmark CROSSCODEEVAL with popular public and proprietary large language models.

**CodeGen** (Nijkamp et al., 2023b,a) is a series of generative code LMs. CodeGen supports left-only context. CodeGen2.5 notably supports fill-in-the-middle and further improves the performance via multi-epoch training. We benchmarked CodeGen models at various sizes from 350M to 16B.

**StarCoder** (Li et al., 2023) is a generative multi-query-based code LM with 15.5B model parameters trained on The Stack dataset (Kocetkov et al., 2022). It supports up to 8k tokens. We also benchmarked its base version with at varied sizes: 1B, 3B, and 7B.

---

[5]Mono is an open-source implementation of the .NET Framework: https://www.mono-project.com/

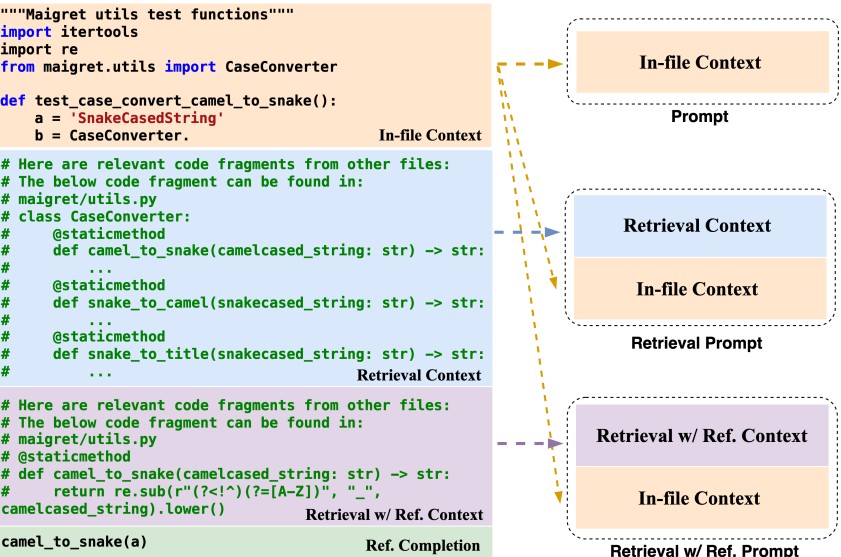

Figure 3: An illustrative example to showcase the use of in-file context, retrieved cross-file context, and retrieved context using reference in prompts. While the baseline prompting uses in-file context only, "Retrieval" and "Retrieval w/ Ref." prompting uses retrieved contexts by prepending them to the in-file context.

**GPT-3.5-turbo** (Ouyang et al., 2022) is one of the most powerful models developed by OpenAI. It was trained with comprehensive text and code data and supports up to 4k max sequence length. Its model weight remains proprietary and is accessible exclusively via APIs.

## 3.2 Evaluation Metrics

In evaluating the performance of code language models, we report performance in two main categories: code match and identifier match (Ding et al., 2022).

**Code Match** The code match metric directly compares the generated code with the reference and is measured using exact match (EM) and edit similarity (ES). These metrics help to assess the overall accuracy of the code completion process, taking into account elements such as identifiers, keywords, operators, delimiters, and literals.

**Identifier Match** This metric evaluates the model's ability to predict the correct application programming interfaces (APIs). To perform this evaluation, we first parse the code and extract the identifiers from the model prediction and reference, resulting in two ordered lists of identifiers. We then compare the predicted identifiers with the reference and report the results in EM and F1 score.

## 3.3 Experimental Setup

Our evaluation framework is based on the Transformers (Wolf et al., 2020) library. All the experiments are conducted with the zero-shot setting, and no training is involved. We use the same set of hyperparameters for code generation across all models. We set the maximum sequence length to 2,048 for the CodeGen family, 4096 for GPT-3.5-turbo, and 8,192 for the StarCoder family. We use a maximum generation length of 50 and the rest as the prompt.

We explore greedy search and nucleus sampling (Holtzman et al., 2020) with reranking (Hossain et al., 2020). We found there is no significant difference between the two, and we present the greedy search results in the main paper and refer readers to Appendix D.2 for nucleus sampling.

We post-process model predictions to extract statements.[6] For Python, we iteratively parse the concatenation of prompt and $n$ completion tokens (e.g., $n = 1, 2, \ldots, 50$) until the sequence becomes

---

[6]We apply the same post-processing on the references before calculating the evaluation metrics.

parseable (no syntax error) and the $(n + 1)$-th completion token is a newline character.[7] For Java, TypeScript, and C#, we consider statements ending with ";", "{" and "}", instead of a new line.

**Only In-File Context**    In standard practice, pre-trained language models are utilized to perform code completion in a *zero-shot* manner by taking into account the provided code context. Following the practice, we conduct experiments using the code LMs (3.1), where they are provided code context from the current file. As shown in Figure 3, the baseline prompt includes only in-file context.

**Retreived Cross-file Context**    Inspired by the effectiveness of the recently proposed retrieve-and-generate (RG) framework for repository-level code completion (Zhang et al., 2023), we adopt it for cross-file context retrieval.[8] In the RG framework, the retrieval database is constructed by iteratively scanning the files in the repository and extracting contiguous $M$ lines (in all our experiments, $M = 10$) of non-overlapping code fragments, which are the candidates for cross-file context retrieval. The query for the retrieval is built using the last $N$ lines (we set $N = 10$) of the in-file context. We use BM25 (Robertson et al., 2009) to calculate the similarity between the query and the candidates (cross-file context chunks), and use the top-5 similar code snippets as the cross-file context, see "Retrieval Context" in Figure 3. We consider a maximum of 512 BPE tokens for such context, and the rest of the tokens will be truncated. Figure 3 illustrates[9] the retrieved context and the corresponding prompt for the model to complete. Given the in-file context as a query, the RG framework successfully retrieves the class definition of `CaseConverter` that is in another file for utilities. We further wrap the class definition into a template as code comment and use it as the cross-file context. To build the retrieval prompt, we prepend the retrieved context to the in-file context.

**Retrieval with Reference**    To quantify the upper bound impacts of cross-file context retrieved by the RG framework, we devise "retrieval with reference" for comparison. In this setting, we make use of not only the in-file context (as in standard retrieval setting) but also *the reference* to retrieve the cross-file context. Specifically, the query is constructed by using the last $N$ lines of *the concatenation of the in-file context and the reference completion*, instead of the in-file context only in the standard retrieval setting. We prepend the retrieved context (*i.e.,* "Retrieval w/ Ref. Context") to in-file context to construct the prompt for this setting.

Note that the Retrieval w/ Ref. context could not be applied to the realistic code completion, as the reference completion is unknown. We use it as an estimation of the upper bound model performance with the RG framework. Also, the model's performance in this setting is not optimal, as it can still be limited by imperfect retrieval and the model's capability in making use of retrieved code, and we perform additional benchmarking and analysis on retrieval methods later in §3.5.

## 3.4   Results

We present results in Table 2 and additional results in Table 7. We see that all models perform poorly when the prompt includes only the in-file context. For example, the best-performing StarCoder model at 15.5B size only reports 8.82% code exact match in Python. Even a large code LM struggles to achieve promising performance in completing CROSSCODEEVAL samples with only in-file context since it could not provide sufficient clues for code completion. This shows the design of CROSSCODEEVAL that cross-file context is necessary to complete the code correctly.

The performance improves dramatically when the cross-file context is added to the prompts across all models and sizes. Figure 4 shows the significant improvements resulting from the inclusion of cross-file context in CodeGen and StarCoder models. Looking at Table 2, we see that the StarCoder model reports up to $3.0\times$ and $4.5\times$ better exact code match when including retrieved and retrieved with reference context respectively. The results underline the limitation of existing datasets that only consider the in-file context to evaluate code LMs, making these datasets insufficient in reflecting models' best capacity in real-world scenarios. In contrast, CROSSCODEEVAL maintains the cross-file

---

[7]We manually verified that this is required to extract full statements in Python.

[8]We tried to use the code made publicly available by the authors of Zhang et al. (2023) but failed to execute them with the provided instructions. As a result, we implemented the RG approach by ourselves. Note that, unlike Zhang et al. (2023), we perform only 1-step retrieval augmented generation.

[9]For the convenience of illustration, we remove the irrelevant code snippets from the retrieved context to avoid confusion. In practice, the retrieved context spans across a fixed length of lines, which includes useful information as well as its surrounding lines. More implementation details are in Appendix C.

contexts for code completion samples, providing resources to both identify the model's best capacity and analyze the model's behavior when seeing a more comprehensive context.

| Model | Code Match | | | | | | | |
| --- | --- | --- | --- | --- | --- | --- | --- | --- |
| | Python | | Java | | TypeScript | | C# | |
| | EM | ES | EM | ES | EM | ES | EM | ES |
| CodeGen25-7B | 7.73 | 59.34 | 10.43 | 62.05 | 7.81 | 57.56 | 4.36 | 58.99 |
| + Retrieval | 14.52 | 64.40 | 16.88 | 64.35 | 12.57 | 60.08 | 13.01 | 63.86 |
| + Retrieval w/ Ref. | 19.17 | 67.46 | 20.20 | 66.17 | 15.35 | 62.73 | 17.87 | 66.14 |
| StarCoder-15.5B | 8.82 | 61.08 | 9.96 | 63.25 | 6.35 | 51.22 | 4.47 | 59.80 |
| + Retrieval | 15.72 | 66.28 | 17.48 | 66.10 | 8.31 | 44.87 | 13.57 | 65.00 |
| + Retrieval w/ Ref. | 21.01 | 68.66 | 19.92 | 67.75 | 11.02 | 46.67 | 20.08 | 67.97 |
| GPT-3.5-turbo | 4.88 | 52.58 | 12.30 | 63.52 | 6.38 | 53.78 | 3.56 | 56.48 |
| + Retrieval | 10.77 | 54.92 | 19.12 | 65.61 | 10.94 | 55.83 | 11.82 | 62.40 |
| + Retrieval w/ Ref. | 15.72 | 58.88 | 22.72 | 68.50 | 14.15 | 58.40 | 17.65 | 66.07 |

| Model | Identifier Match | | | | | | | |
| --- | --- | --- | --- | --- | --- | --- | --- | --- |
| | Python | | Java | | TypeScript | | C# | |
| | EM | F1 | EM | F1 | EM | F1 | EM | F1 |
| CodeGen25-7B | 14.26 | 46.02 | 16.60 | 51.43 | 12.46 | 47.75 | 7.69 | 33.81 |
| + Retrieval | 22.96 | 53.68 | 24.03 | 55.48 | 17.85 | 51.27 | 17.36 | 43.56 |
| + Retrieval w/ Ref. | 28.33 | 57.95 | 27.91 | 57.87 | 21.51 | 55.38 | 21.78 | 47.63 |
| StarCoder-15.5B | 15.72 | 48.16 | 18.28 | 53.23 | 11.86 | 43.53 | 8.54 | 34.33 |
| + Retrieval | 24.77 | 55.57 | 25.95 | 57.74 | 14.09 | 39.50 | 18.04 | 44.38 |
| + Retrieval w/ Ref. | 30.24 | 59.46 | 29.73 | 60.47 | 17.55 | 42.18 | 24.38 | 49.09 |
| GPT-3.5-turbo | 10.09 | 39.18 | 18.93 | 52.52 | 10.76 | 44.78 | 5.77 | 30.25 |
| + Retrieval | 17.37 | 44.43 | 26.74 | 56.57 | 16.69 | 48.15 | 15.44 | 41.24 |
| + Retrieval w/ Ref. | 23.49 | 50.14 | 31.79 | 60.52 | 20.65 | 51.54 | 21.72 | 47.21 |

Table 2: Performance of various code LMs on CROSSCODEEVAL."Retrieval" and "Retrieval w/ Ref." mean we construct the prompt by prepending the retrieved cross-file context retrieved with the prompt and the prompt + reference (see §3.3 for details). The performance with no cross-file context (first row in each section) is generally poor. When prompts are augmented with cross-file context (middle row in each section), the performance increases significantly. The use of reference completion in formulating the query for cross-file context retrieval (last row in each section) shows the upper bound of the retrieve-and-generate (RG) approach. Results of other models are in Table 7.

## 3.5 Analysis and Discussions

**Improved vs. Degraded Code Completions**  Table 3 presents changes in the number of correct completions (based on exact match to the references) across different prompt settings. The results suggest that all models follow a trend that the performance improves with better cross-file context (In-file → Retrieval → Retrieval w/ Ref.). However, the variation of correct/incorrect generation is significant; for example, when changing from Retrieval to Retrieval w/ Ref. with StarCoder in CROSSCODEEVAL Python, we see 327 correct generations changed to incorrect, and 468 generations changed the other way around. Upon manual inspections, we see that the retrieval of the correct cross-file context plays a huge role, as the quality of the retrieval directly correlates with whether the model is able to generate correctly. This effect is further enhanced by the fact that the retrieval happens in fixed lines of code that do not often follow code structure, making it difficult for the model to digest, especially at zero-shot settings, echoing results from Zhang et al. (2023). This highlights that the current best models are still imperfect in leveraging extensive context to make better code completion. Further, it calls for additional studies in optimizing the retrieval methods for code: we show benchmarking with various retrieval methods later in the section.

| Model | Python | | | | | Java | | | | |
|---|---|---|---|---|---|---|---|---|---|---|
| | In-file | → | Ret. | → | Ret. w/ Ref | In-file | → | Ret. | → | Ret. w/ Ref |
| CodeGen-350M | +72 | $-68$ $+182$ | +186 | $-166$ $+237$ | +257 | +75 | $-71$ $+145$ | +149 | $-135$ $+158$ | +172 |
| CodeGen-16.1B | +183 | $-162$ $+311$ | +332 | $-259$ $+371$ | +444 | +150 | $-133$ $+233$ | +250 | $-209$ $+251$ | +292 |
| CodeGen25-7B | +206 | $-172$ $+353$ | +387 | $-306$ $+430$ | +511 | +223 | $-179$ $+317$ | +361 | $-277$ $+348$ | +432 |
| StarCoder | +235 | $-192$ $+376$ | +419 | $-327$ $+468$ | +560 | +213 | $-167$ $+328$ | +374 | $-300$ $+352$ | +426 |

| Model | TypeScript | | | | | C# | | | | |
|---|---|---|---|---|---|---|---|---|---|---|
| | In-file | → | Ret. | → | Ret. w/ Ref | In-file | → | Ret. | → | Ret. w/ Ref |
| CodeGen-350M | +93 | $-85$ $+127$ | +135 | $-128$ $+168$ | +175 | +16 | $-16$ $+58$ | +58 | $-51$ $+108$ | +115 |
| CodeGen-16.1B | +151 | $-140$ $+226$ | +237 | $-218$ $+289$ | +308 | +32 | $-29$ $+95$ | +98 | $-85$ $+138$ | +151 |
| CodeGen25-7B | +262 | $-226$ $+386$ | +422 | $-341$ $+434$ | +515 | +77 | $-62$ $+215$ | +230 | $-189$ $+275$ | +316 |
| StarCoder | +213 | $-198$ $+264$ | +279 | $-241$ $+338$ | +376 | +79 | $-66$ $+227$ | +240 | $-195$ $+310$ | +355 |

Table 3: The numbers of correct code completions using different code generation models on the CROSSCODEEVAL benchmark. "In-file" refers to the prompts being constructed only with in-file context, and "Retrieval" and "Ret. w/ Ref" refer to the prompts being constructed with the retrieved contexts described in §3.3.

**Scalability of Model Performance** Figure 4 visualizes how the performance of CodeGen and StarCoder scales w.r.t. model sizes. We see the performance increases following the power law in all settings as expected (Kaplan et al., 2020). However, again the performance is far from perfect even with the best performing model with the best context retrieved.

**Locations of Retrieved Cross-file Context** To identify relevant cross-file context, we retrieve code snippets from other files of the repository. To understand which files contribute to the cross-file context, we further conduct a study on the retrieved code snippets. To analyze the code snippets retrieved for each prompt, we examine the files to determine whether they meet the following criteria: (1) they are imported by the target file, (2) they are located in the same directory as the target file, (3) they have a similar name to

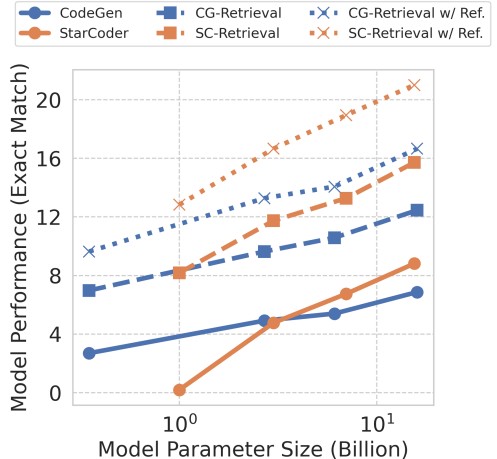

Figure 4: Performance of models in various sizes.

the target file (with filename sharing at least one token, assuming snake-case or CamelCase style filenames), and (4) they include at least one API import within the project, similar to the target file. Our analysis shows that most of the code snippets are sourced from files that are either from the same directory with the target file (Python: 49.0%, Java: 37.8%, TypeScript: 51.3%, C#: 51.7%), or have similar names (Python:33.4%, Java:44.5%, TypeScript: 24.9%, C#: 39%). We also observed that target files and cross-files often share at least one intra-project API import statement. This result aligns with the findings of Zhang et al. (2023).

**Identifier Overlap with Retrieved Cross-file Context** Identifiers are a significant part of programming language constructs that cover API mentions in a source code. Therefore, we examine the distribution of the retrieved cross-file contexts for examples in CROSSCODEEVAL that include mentions of identifiers that are also present in the references. In Figure 5, we show the distribution and the identifier exact match performance achieved by the best performing code LM, StarCoder. In general, it is evident that an increased ratio of identifier overlap results in higher performance,

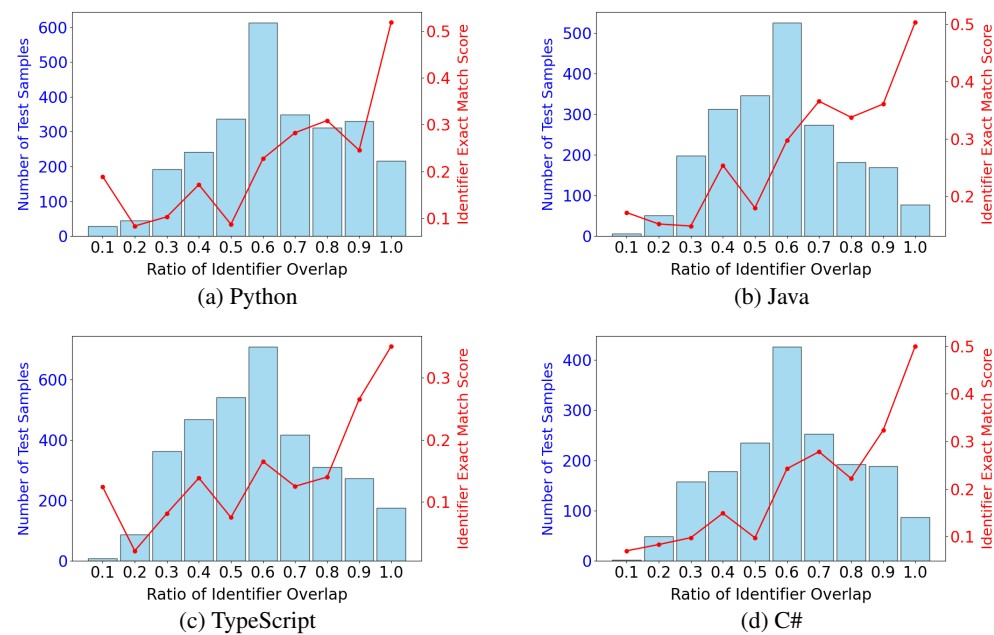

|        | (a) Python | (b) Java |
|--------|------------|----------|
|        | (c) TypeScript | (d) C# |

Figure 5: Distribution of the examples according to identifier overlap between the retrieved cross-file context and the reference completion. We also show the corresponding identifier exact match scores.

| Model | Retriever | Code Match | | | | | | | |
|-------|-----------|------------|---|------|---|-----------|---|-----|---|
|       |           | **Python** | | **Java** | | **TypeScript** | | **C#** | |
|       |           | EM | ES | EM | ES | EM | ES | EM | ES |
| CodeGen25-7B | - | 7.73 | 59.34 | 10.43 | 62.05 | 7.81 | 57.56 | 4.36 | 58.99 |
| + Retrieval | BM25 | 14.52 | 64.40 | 16.88 | 64.35 | 12.57 | 60.08 | 13.01 | 63.86 |
| + Retrieval w/ Ref. | BM25 | 19.17 | 67.46 | 20.20 | 66.17 | 15.35 | 62.73 | 17.87 | 66.14 |
| + Retrieval | UniXCoder | 13.73 | 64.21 | 15.61 | 63.67 | 12.10 | 59.82 | 12.39 | 63.82 |
| + Retrieval w/ Ref. | UniXCoder | 18.01 | 66.46 | 18.19 | 65.23 | 14.84 | 61.66 | 16.46 | 65.20 |
| + Retrieval | OpenAI ada | 14.82 | 65.00 | 17.77 | 64.48 | 12.75 | 60.02 | 14.71 | 65.35 |
| + Retrieval w/ Ref. | OpenAI ada | 18.39 | 66.80 | 20.94 | 66.27 | 15.58 | 62.65 | 20.43 | 68.65 |
| StarCoder-15.5B | - | 8.82 | 61.08 | 9.96 | 63.25 | 6.35 | 51.22 | 4.47 | 59.80 |
| + Retrieval | BM25 | 15.72 | 66.28 | 17.48 | 66.10 | 8.31 | 44.87 | 13.57 | 65.00 |
| + Retrieval w/ Ref. | BM25 | 21.01 | 68.66 | 19.92 | 67.75 | 11.02 | 46.67 | 20.08 | 67.97 |
| + Retrieval | UniXCoder | 15.87 | 66.07 | 16.83 | 66.09 | 7.87 | 44.67 | 11.93 | 63.90 |
| + Retrieval w/ Ref. | UniXCoder | 19.32 | 68.33 | 19.45 | 67.51 | 10.28 | 46.85 | 16.63 | 66.30 |
| + Retrieval | OpenAI ada | 16.47 | 66.72 | 17.53 | 65.98 | 8.43 | 45.08 | 15.39 | 66.21 |
| + Retrieval w/ Ref. | OpenAI ada | 20.53 | 68.50 | 21.69 | 68.11 | 11.83 | 47.31 | 23.49 | 70.56 |

Table 4: Evaluation results of various sparse and neural methods in retrieving cross-file context. Identifier Match results are in Table 9.

demonstrating a positive correlation. This calls for an investigation into retrieval techniques, with a particular emphasis on key terms like identifiers for cross-file context retrieval.

**CROSSCODEEVAL as Code Retrieval Benchmark** The observations above (e.g., imperfect upper bound performance and identifier overlap) underscore the critical role of the code retrieval method. Given the strong dependency that the correct prediction requires an accurate retrieval of relevant cross-file context, we propose to use CROSSCODEEVAL as a code retrieval benchmark. We perform experiments with different retrievers from sparse (BM25 as we used in the rest of the experiments) to neural (UniXCoder (Guo et al., 2022) and OpenAI embedding[10]). For UniXCoder, we use a max

---

[10]We use `text-embedding-ada-002`.

sequence length of 256 per 10-line chunk and for OpenAI embedding we use 8,000. We use the cosine similarity of the embedding of the prompt and the chunks to retrieve top 5 chunks.

Table 4 shows the results with these retrieval methods. On one hand, we see BM25 provides a strong baseline and, in most of the cases, can outperform UniXCoder-based retriever. On the other hand, retrieving with OpenAI's ada embedding is generally better than both BM25 and UniXCoder, especially for Java and C#. Nonetheless, the performance with the best performing retriever is still suboptimal ($< 20$ EM in all languages), calling for future development of better code retriever.

## 4  Related Works

The advent of code language models (LMs) (Feng et al., 2020; Ahmad et al., 2021; Wang et al., 2021; Guo et al., 2022) have bolstered the automation of software engineering applications. Among them, code completion has got the most attention, and as a result, generative AI powered by large language models for code (Chen et al., 2021; Xu et al., 2022; Wang and Komatsuzaki, 2021; Black et al., 2021, 2022; Nijkamp et al., 2023b; Fried et al., 2023; Li et al., 2022; CodeGeeX, 2022; Allal et al., 2023; Li et al., 2023; Nijkamp et al., 2023a) has become a reality. Benchmark datasets have been playing a pivotal role in advancing the field of generative AI for code. A large pool of recent works (Chen et al., 2021; CodeGeeX, 2022; Austin et al., 2021; Athiwaratkun et al., 2023; Cassano et al., 2023; Hendrycks et al., 2021; Raychev et al., 2016; Lu et al., 2021; Allamanis and Sutton, 2013; Puri et al., 2021; Husain et al., 2019; Clement et al., 2021; Ding et al., 2023; Wang et al., 2023; Lu et al., 2022) developed benchmarks to facilitate the evaluation of code LMs. These benchmarks typically assess code completion ability given in-file context – code prompts containing code snippets from current files (where the user is writing code). Therefore, the capability of these code LMs to generate code that requires software repository-level context has been left unexplored until recently.

A few recent works proposed repository-level code generation frameworks and benchmarks (Shrivastava et al., 2023; Ding et al., 2022; Pei et al., 2023; Zhang et al., 2023). While these works share high-level insights with CROSSCODEEVAL, highlighting the importance of cross-file context, their focus is mainly on proposing a new approach to incorporate such contexts, and datasets are collected to evaluate their own approaches rather than being carefully crafted as a benchmark to evaluate the code LMs in general. For example, Shrivastava et al. (2023) and Ding et al. (2022) only collect data for one single programming language, and Pei et al. (2023) narrows the completion scope to only function arguments. As a comparison, CROSSCODEEVAL comprehensively includes four different programming languages (Python, Java, Typescript, and C#) and targets evaluating the general code completion capacity of code LMs rather than a specific type of application. REPOEVAL (Zhang et al., 2023) is a concurrent work building repository-level code completion benchmark in Python, constructed from 16 GitHub repositories. These repositories are limited in a domain (mainly academia/research work), some of them overlap with popular code pre-training datasets (such as The Stack (Kocetkov et al., 2022)), and some are with non-permissive licenses. In contrast, CROSSCODEEVAL is derived from a diverse pool of permissively licensed GitHub repositories in 4 popular languages (§2.4). Furthermore, CROSSCODEEVAL does not overlap with The Stack to avoid data leakage, minimizing potential memorization issues during evaluations.

## 5  Conclusion

We introduce CROSSCODEEVAL, a diverse and multilingual benchmark for cross-file code completion. CROSSCODEEVAL necessitates cross-file contextual understanding to complete the code accurately. We use a static-analysis-based method to identify cross-file context usages in code, and take steps to ensure the dataset is of high quality and has minimal data leakage with the pre-training dataset of popular code LMs. We experiment with popular code language models and results show that the inclusion of cross-file context significantly improves their accuracy in code completion, demonstrating that CROSSCODEEVAL is an effective benchmark assessing cross-file code completion capabilities. Moreover, even the top-performing model with the best retrieval method still exhibits great room for improvement, highlighting the need for further advancements in leveraging extensive context for code completion and better code retriever. In both directions, CROSSCODEEVAL stands as a pivotal benchmark. We envision CROSSCODEEVAL could fill in the gap of evaluating code completion that requires cross-file context and promote future research in all dimensions in this direction.

## Acknowledgments

We would like to thank Ramana Keerthi and Akhilesh Bontala for their help in data collection, and Bryan McWhorter for various legal consultations.

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

# CROSSCODEEVAL: A Diverse and Multilingual Benchmark for Cross-File Code Completion

# Appendices

## A More Details of CROSSCODEEVAL Generation

### A.1 Python

We presented a Python example in Figure 2, Section 2.2 in the main paper. Here we describe the static analysis part in detail. Recall that the purpose of static analysis is to detect undefined members in the modified file whose imports are replaced by empty classes. Hence, there's no need to include project-level information at this stage, and we only run Pylint on the standalone file. We restrict the output types to be error only and ignore other types such as warnings and coding conventions. In the example in Figure 2, Section 2.2, the Pylint output for the modified file is "`test_utils.py:10:8:  E1101: Class 'CaseConverter' has no 'camel_to_snake' member (no-member)`". We collect all the errors of type `E1101` and use them to identify locations of cross-file context usage.

### A.2 Java

Instead of using static analysis tool, we directly leverage Java default compiler to extract the undefined member functions in the modified file, whose belonging class, originally imported from another project file, is replaced by an empty class. Figure 6 shows the work flow of the cross-file method extraction. Specifically, we can parse the new errors with the ∧ mark in the modified code to locate the calls of the cross-file function. An illustration is provided in Figure 7.

### A.3 TypeScript

We go through the following steps to create examples from TypeScript repositories.

1. Extract import statements from source files (files with ".ts" or ".tsx" extension) and identify the dependencies. For example, a file `fileA.ts` having the import statement `import module_name from '../fileB.ts';` depends on `../fileB.ts`. (`../fileB.ts` is a cross-file for `fileA.ts`)

2. Identify the *cross-file lines* by modifying the source code, followed by compiling the project using `tsc`.[11] We make the following two types of modifications.

   - Removing the import statement and checking the compilation output message for "`error TS2304:  Cannot find name 'X'.`".
   - Removing the import statement and adding a dummy class "`class module_name {}`" and checking the compilation output message for "`error TS2339:  Property 'X' does not exist on type 'Y'.`".

3. From the compiler error message, we use the line numbers to form references for examples.

### A.4 C#

For C# repositories, the example generation procedure follows two steps. In the first step, we pick a source code file from a C# repository and replace the original class with a dummy one. Then, we run the C# mono compiler on all C# files and gather the compilation errors. In the next step, we compare the compiler error messages before and after the substitution in step-1 to locate the cross-file references based on the incremental errors, and extract the line numbers for references in CROSSCODEEVAL examples. We repeat the process for each source code file within a repository.

---

[11]https://www.typescriptlang.org/docs/handbook/compiler-options.html

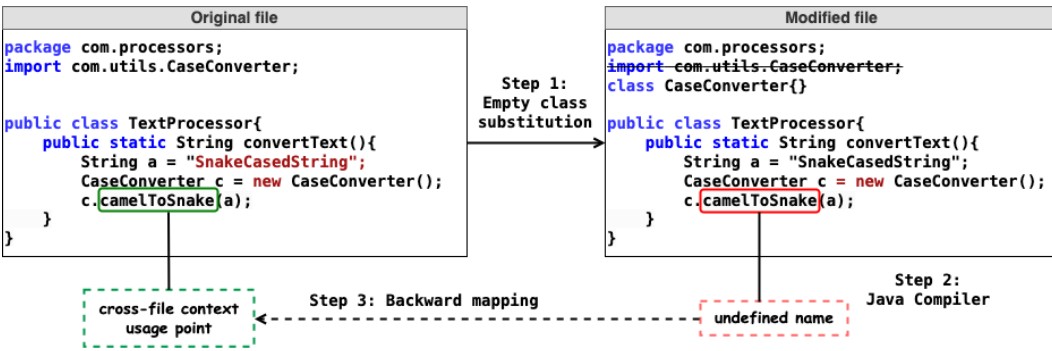

Figure 6: Java data creation: 1) given a original code file, we create an adapted version by replacing an imported class found in the project directory and generating a dummy class accordingly. 2) Run *javac* on both versions. New *javac* error points of the modified code are actually the method calls of the original code, where the definitions are in another file in the project. A pair of prompt and ground truth completion is generated accordingly.

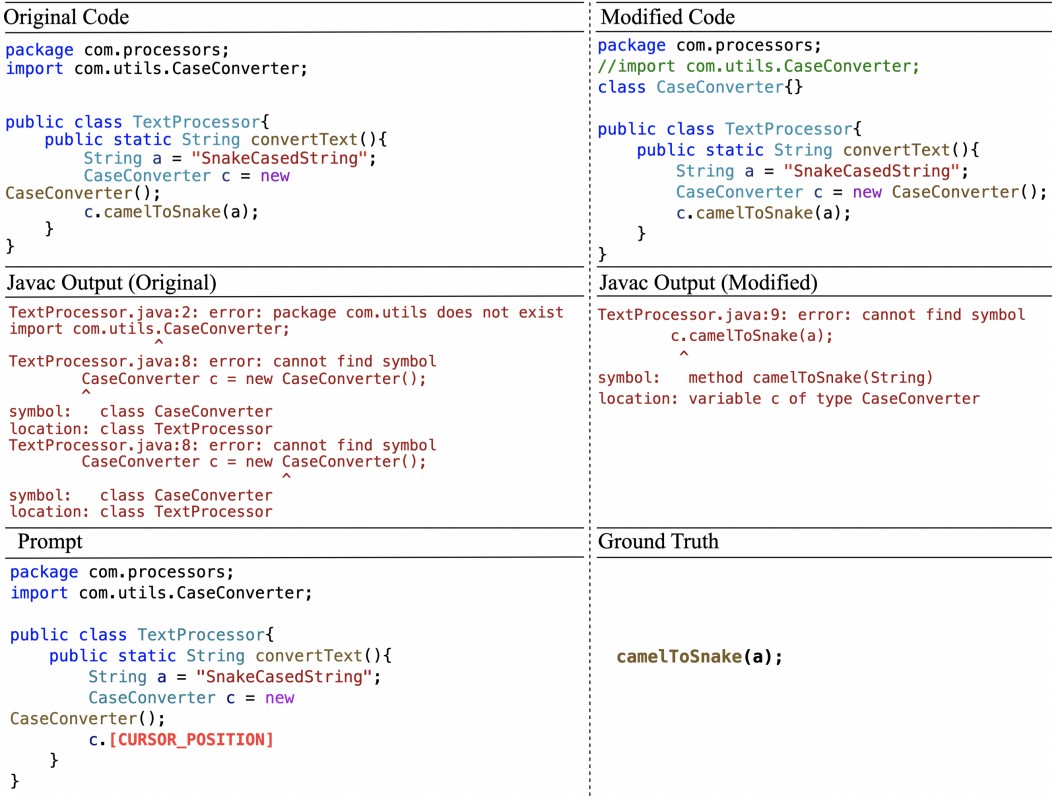

Figure 7: By comparing *javac* errors of original Java code and modified code, we find the cross-file entities and create the prompt and ground truth.

## B   Human Annotation for Quality Control

To assess the quality and further improve CROSSCODEEVAL, we conducted human annotation. We randomly sampled 100 and 50 examples from CROSSCODEEVAL Python and Java sets, respectively. Six authors annotated 50 example each, and each example was annotated by two annotators with significant experience in the targeted language. Three questions were asked in the annotation:

1. Does the reference contain any name that requires looking at the associated cross-file? [Pay attention to: while answering, think if looking at the associated cross-file helps you knowing

more info about a name mentioned in the reference.]
Choose from A: Yes. / B: No, I already know everything about all the names mentioned in the reference. / C: Reference does not contain any such name.

2. Can you predict the reference given only the current-file context? [Pay attention to: while answering, think if you can guess the reference given the current-file context.]
Choose from A: No, I cannot predict. / B: I possibly can predict but it may not exactly match the reference. / C: I can predict the reference.

3. Will you prefer to remove the example from the dataset? [Pay attention to: Is the example good enough to be included in the dataset? Use your best judgment to make a decision.]
Choose from A: No. / B: Maybe. / C: Yes.

|        | Q1   | Q2  | Q3  |
|--------|------|-----|-----|
| Python | 98%  | 89% | 88% |
| Java   | 100% | 84% | 76% |

Table 5: Agreement scores for the annotation questions in Python and Java, respectively.

|        | Q1   |    |    | Q2    |      |    | Q3    |     |      |
|--------|------|----|----|-------|------|----|-------|-----|------|
|        | A    | B  | C  | A     | B    | C  | A     | B   | C    |
| Python | 99%  | 1% | 0% | 88.5% | 9.5% | 2% | 91.5% | 6%  | 2.5% |
| Java   | 100% | 0% | 0% | 88%   | 10%  | 2% | 80%   | 13% | 7%   |

Table 6: Distribution of annotation answers to each question in Python and Java.

We calculated the agreement score[12] and summarize the distribution of the annotations in Tables 5 & 6. Overall, we see great agreement scores in most of the questions in both languages, suggesting annotators have consensus in these questions. Looking at the distribution of annotation answers, we see that in almost 100% cases, the references contain names that necessities cross-file information (Q1), and only at 2% that the reference can be predicted with only current-file context (Q2). Both together suggest that CROSSCODEEVAL serves its purpose to be a dedicated cross-file code completion benchmark that accurately reflects model's capability in cross-file context understanding. Besides, we see 2.5% and 7% examples that annotators think should be removed from the dataset (Q3). A closer look reveals that many of such examples contain long strings in the reference, e.g., `Chat.sendClientSystemMessage("Available scripts:");`, which cannot be predicted easily and also led to disagreement between annotators. We plan to improve the dataset by filtering out examples with long string in the reference in the next data revision.

## C  Retrieve-and-Generate Modeling Details

We present Figure 8 to illustrate more details regarding the Retrieve-and-Generate (RG) framework.

**Cross-file Context Retrieval**    As we have introduced in Section 3.3, the query will be constructed using the in-file context, i.e., the last 10 code lines before the cursor position, and the goal is to find the relevant cross-file context – a similar operation of the deposit that has been implemented in the class `Bank`. The code snippets in other files are chunked as code snippets with 10 lines as candidates for similarity calculation, and the most similar candidate of the in-file context turns out to be lines 5-14 in "Relevant Cross-file Context" of Figure 8. Since the in-file context is left context and the target completion is the following code, correspondingly, the retrieved context will be *the following lines of the similar candidate, i.e., line 15-23 in "Relevant Cross-file Context"*. We can see that line 15 gives a direct hint that helps with the prediction.

**Cross-file Context Retrieved w/ Reference**    As an upper bound estimate, the query for the oracle context will be constructed by the last 10 lines of concatenating the in-file context and the target completion. Consequently, the most similar candidate is lines 8-17 in "Relevant Cross-file Context",

---

[12]Given we only have limited examples and annotations per example, and the annotation is relatively sparse, we didn't include other inter-coder agreement metrics like Krippendorff's $\alpha$ or Cohen's $\kappa$.

| In-file Context |
|---|

```python
from getpass import getpass
import SQL_Control_Functions as SCF

def BankBasedOptions(acc_no:int):
    while True:
        print("Select an option from below")
        print("1. Deposit")
        print("2. Withdraw")
        print("3. Show Details")
        print("4. Show Statements")
        print("5. Send Money")
        print("6. Exit")
        choice = int(input("Enter your choice: "))
        if choice == 1:
            amount = int(input("Enter amount to be deposited: "))
            [CURSOR_POSITION]
```

| Target Completion |
|---|

```python
SCF.Transaction(mydb, acc_no, amount)
```

| Relevant Cross-file Context |
|---|

```python
0   # Local File: ./Program.py
1   class Bank:
2       acc_count = int(0)
3       def __init__(self, name, balance):
4           pass
5       def BankBasedOptions(acc_no:int):
6           while True:
7               print("Select an option from below")
8               print("1. Deposit")
9               print("2. Withdraw")
10              print("3. Show Details")
11              print("4. Exit")
12              choice = int(input("Enter your choice: "))
13              if choice == 1:
14                  amount = int(input("Enter amount to be deposited: "))
15                  SCF.Transaction(mydb, acc_no, amount)
16              elif choice == 2:
17                  amount = int(input("Enter amount to be withdrawn: "))
18                  Bank.withdraw(mydb, acc_no, -amount)
19              elif choice == 3:
20                  SCF.PrintTable(mydb, "Account", acc_no)
21              elif choice == 4:
22                  print("\n\n")
23                  Bank.EntryOptions()
```

Figure 8: An illustration demonstrating the context retrieval.

which, different from the cross-file context, will be directly used as the oracle context, since the query already contains the target completion rather than just left context. Again, it includes the direct hint, line 15, for the code completion.

# D   Additional Evaluation Results and Ablations

## D.1   Evaluation with Additional Models

Beyond the results in Table 2 of the main paper, we further evaluate more code LMs, including more variants of CodeGen and StarcoderBase. The results are shown in Table 7.

| Model | Code Match | | | | | | | |
| | Python | | Java | | TypeScript | | C# | |
| | EM | ES | EM | ES | EM | ES | EM | ES |
|---|---|---|---|---|---|---|---|---|
| CodeGen-350M | 2.70 | 52.94 | 3.51 | 54.92 | 2.77 | 46.60 | 0.90 | 42.64 |
| + Retrieval | 6.98 | 56.91 | 6.97 | 56.87 | 4.02 | 48.07 | 3.28 | 45.35 |
| + Retrieval w/ Ref. | 9.64 | 59.31 | 8.04 | 57.75 | 5.21 | 50.25 | 6.50 | 48.58 |
| CodeGen-2.7B | 4.92 | 55.75 | 5.89 | 58.86 | 3.84 | 50.77 | 1.36 | 48.04 |
| + Retrieval | 9.64 | 60.02 | 10.00 | 61.20 | 6.47 | 52.61 | 4.24 | 52.19 |
| + Retrieval w/ Ref. | 13.28 | 62.65 | 11.73 | 62.50 | 7.81 | 54.79 | 7.35 | 54.78 |
| CodeGen-6.1B | 5.40 | 55.41 | 5.19 | 58.89 | 3.75 | 49.87 | 1.47 | 45.34 |
| + Retrieval | 10.58 | 60.36 | 8.88 | 61.06 | 6.14 | 51.58 | 4.64 | 47.53 |
| + Retrieval w/ Ref. | 14.07 | 62.80 | 11.03 | 61.76 | 8.22 | 54.01 | 7.13 | 48.98 |
| CodeGen-16.1B | 6.87 | 57.64 | 7.01 | 60.49 | 4.50 | 52.24 | 1.81 | 45.28 |
| + Retrieval | 12.46 | 62.66 | 11.69 | 62.16 | 7.06 | 54.11 | 5.54 | 48.25 |
| + Retrieval w/ Ref. | 16.66 | 65.31 | 13.65 | 63.50 | 9.18 | 56.54 | 8.54 | 49.63 |
| StarCoderBase-1B | 0.19 | 54.78 | 0.19 | 57.17 | 0.06 | 42.00 | 0.11 | 57.65 |
| + Retrieval | 8.18 | 60.06 | 7.48 | 59.71 | 3.64 | 40.07 | 8.94 | 62.32 |
| + Retrieval w/ Ref. | 12.83 | 63.37 | 10.38 | 62.06 | 6.79 | 42.47 | 15.16 | 65.56 |
| StarCoderBase-3B | 4.77 | 57.54 | 5.66 | 60.26 | 3.55 | 46.74 | 3.62 | 59.27 |
| + Retrieval | 11.74 | 62.99 | 12.39 | 62.33 | 6.73 | 43.89 | 11.99 | 64.65 |
| + Retrieval w/ Ref. | 16.66 | 65.91 | 14.91 | 64.83 | 9.62 | 46.39 | 18.04 | 66.86 |
| StarCoderBase-7B | 6.75 | 59.83 | 8.74 | 62.84 | 5.13 | 49.31 | 4.86 | 59.71 |
| + Retrieval | 13.28 | 64.76 | 15.61 | 65.19 | 8.25 | 45.73 | 14.20 | 65.65 |
| + Retrieval w/ Ref. | 18.95 | 67.41 | 18.70 | 67.76 | 11.95 | 47.91 | 20.64 | 67.95 |

| Model | Identifier Match | | | | | | | |
| | Python | | Java | | TypeScript | | C# | |
| | EM | F1 | EM | F1 | EM | F1 | EM | F1 |
|---|---|---|---|---|---|---|---|---|
| CodeGen-350M | 7.92 | 38.51 | 9.40 | 43.36 | 5.07 | 34.44 | 1.98 | 21.07 |
| + Retrieval | 13.70 | 44.34 | 13.32 | 46.47 | 7.06 | 37.09 | 5.83 | 26.67 |
| + Retrieval w/ Ref. | 17.49 | 48.14 | 14.68 | 47.90 | 8.85 | 39.75 | 8.88 | 31.27 |
| CodeGen-2.7B | 11.14 | 42.42 | 12.62 | 47.94 | 7.51 | 39.71 | 3.96 | 24.92 |
| + Retrieval | 17.22 | 48.96 | 18.19 | 51.43 | 10.91 | 42.96 | 8.03 | 31.85 |
| + Retrieval w/ Ref. | 21.54 | 52.40 | 20.06 | 53.19 | 12.96 | 45.58 | 11.43 | 36.50 |
| CodeGen-6.1B | 10.92 | 42.22 | 12.48 | 48.38 | 6.76 | 39.13 | 3.90 | 23.20 |
| + Retrieval | 17.90 | 49.41 | 17.25 | 51.79 | 10.31 | 42.29 | 8.26 | 30.01 |
| + Retrieval w/ Ref. | 22.40 | 53.14 | 19.45 | 53.12 | 12.87 | 45.72 | 12.10 | 34.20 |
| CodeGen-16.1B | 13.28 | 44.98 | 14.40 | 49.97 | 8.76 | 41.95 | 4.13 | 23.49 |
| + Retrieval | 20.53 | 51.95 | 19.50 | 52.97 | 12.13 | 44.65 | 9.33 | 31.08 |
| + Retrieval w/ Ref. | 25.37 | 55.76 | 22.21 | 54.98 | 15.14 | 48.02 | 12.67 | 35.08 |
| StarCoderBase-1B | 6.79 | 39.70 | 7.71 | 45.18 | 4.71 | 33.58 | 4.02 | 30.32 |
| + Retrieval | 15.91 | 47.87 | 15.57 | 49.64 | 8.13 | 33.45 | 13.01 | 39.38 |
| + Retrieval w/ Ref. | 21.61 | 52.95 | 19.54 | 52.57 | 11.80 | 36.55 | 18.83 | 44.81 |
| StarCoderBase-3B | 11.48 | 43.63 | 14.12 | 49.13 | 8.76 | 39.18 | 7.30 | 32.55 |
| + Retrieval | 19.81 | 51.66 | 21.18 | 53.03 | 11.83 | 37.95 | 16.06 | 42.45 |
| + Retrieval w/ Ref. | 25.40 | 55.89 | 24.36 | 56.09 | 15.08 | 41.09 | 21.95 | 47.03 |
| StarCoderBase-7B | 13.92 | 46.65 | 16.88 | 52.18 | 10.04 | 42.15 | 8.71 | 34.04 |
| + Retrieval | 22.29 | 54.14 | 24.92 | 56.70 | 13.92 | 40.25 | 18.16 | 44.55 |
| + Retrieval w/ Ref. | 28.26 | 58.14 | 29.64 | 59.99 | 17.79 | 43.40 | 24.04 | 49.12 |

Table 7: Evaluation of additional code LMs on CROSSCODEEVAL (cf. Table 2)

## D.2 Nucleus Sampling w/ Re-ranking

Though the main results in this benchmark are reported with greedy search, we further conduct experiments to explore the effects of sampling and reranking. To this end, we apply the nucleus sampling (Holtzman et al., 2020) then mean-log-likelihood reranking (Hossain et al., 2020) during the code generation. The experiments are conducted with temperature of 0.2 for the token probability scaling. For the nucleus sampling, we set the top-p to be 0.95, and for the mean-log-likelihood reranking, we generate top-5 prediction under the sampling setting and compute the mean log-likelihood of each generation to pick the most probable prediction.

The results are shown in Table 8. Compared to the main results (Table 2 in the main paper), we notice the results are quite comparable, and the difference is marginal. The cross-file and oracle context bring equivalent improvement to the greedy search setting. The results empirically reveal that (1) the performance difference of sampling with re-ranking and greedy decoding is marginal, and (2) the cross-file context is helpful regardless of the sampling/search algorithms.

| Model | Code Match | | | | | | | |
| --- | --- | --- | --- | --- | --- | --- | --- | --- |
| | **Python** | | **Java** | | **TypeScript** | | **C#** | |
| | EM | ES | EM | ES | EM | ES | EM | ES |
| CodeGen25-7B | 8.14 | 59.72 | 10.47 | 62.54 | 7.90 | 57.69 | 3.90 | 59.73 |
| + Retrieval | 14.60 | 64.56 | 17.63 | 64.49 | 13.32 | 60.35 | 13.29 | 64.55 |
| + Retrieval w/ Ref. | 19.51 | 67.67 | 20.48 | 66.92 | 15.79 | 62.88 | 18.21 | 66.46 |
| StarCoder-15.5B | 8.93 | 61.43 | 10.66 | 63.97 | 6.05 | 51.95 | 4.64 | 60.52 |
| + Retrieval | 15.68 | 66.70 | 17.58 | 66.73 | 8.76 | 45.78 | 14.03 | 65.70 |
| + Retrieval w/ Ref. | 21.35 | 69.44 | 20.20 | 68.49 | 11.65 | 47.32 | 19.97 | 68.32 |

| Model | Identifier Match | | | | | | | |
| --- | --- | --- | --- | --- | --- | --- | --- | --- |
| | **Python** | | **Java** | | **TypeScript** | | **C#** | |
| | EM | F1 | EM | F1 | EM | F1 | EM | F1 |
| CodeGen25-7B | 14.60 | 46.13 | 17.16 | 52.04 | 12.40 | 47.73 | 7.47 | 34.52 |
| + Retrieval | 22.81 | 53.72 | 25.15 | 55.99 | 18.18 | 51.72 | 17.36 | 43.88 |
| + Retrieval w/ Ref. | 28.78 | 58.20 | 28.94 | 58.98 | 21.99 | 55.53 | 22.12 | 47.78 |
| StarCoder-15.5B | 15.91 | 48.08 | 19.07 | 54.08 | 11.50 | 44.13 | 8.31 | 34.58 |
| + Retrieval | 24.62 | 55.75 | 26.46 | 58.46 | 14.78 | 40.59 | 18.33 | 44.55 |
| + Retrieval w/ Ref. | 30.77 | 60.01 | 30.53 | 61.07 | 18.15 | 42.90 | 24.15 | 49.26 |

Table 8: Performance of code LMs on CROSSCODEEVAL with temperature-based nucleus sampling. (cf. Table 2 in the main body of the paper).

## D.3 Additional Results of Code Retrieval

Table 9 presents identifier match results of various retrieval methods for cross-file context.

## D.4 Qualitative Analysis

To illustrate the quality of the retrieved cross-file context and in which ways they are helping to maximize the code LMs' capacity, we provide two qualitative examples in Figure 9 and 10.

In Figure 9, we can see that the cross-file context provides retrieves the code snippets with a similar context to the cursor position. By capturing the repetitiveness of the repository (Le Goues et al., 2012; Barr et al., 2014), the cross-file context helps the code LMs adapt the existing, repetitive coding patterns to complete the programs. Specifically, the retrieved context also defines a function named step(), with a similar goal of generating a token, and it provides a direct reference for completing the API call.

| Model | Retriever | Identifier Match | | | | | | | |
| | | Python | | Java | | TypeScript | | C# | |
| | | EM | F1 | EM | F1 | EM | F1 | EM | F1 |
| CodeGen25-7B | - | 14.26 | 46.02 | 16.60 | 51.43 | 12.46 | 47.75 | 7.69 | 33.81 |
| + Retrieval | BM25 | 22.96 | 53.68 | 24.03 | 55.48 | 17.85 | 51.27 | 17.36 | 43.56 |
| + Retrieval w/ Ref. | BM25 | 28.33 | 57.95 | 27.91 | 57.87 | 21.51 | 55.38 | 21.78 | 47.63 |
| + Retrieval | UnixCoder | 22.74 | 53.13 | 22.67 | 54.63 | 17.43 | 51.04 | 16.63 | 42.98 |
| + Retrieval w/ Ref. | UnixCoder | 26.90 | 56.53 | 25.57 | 56.69 | 20.74 | 53.79 | 20.36 | 45.74 |
| + Retrieval | OpenAI Ada | 23.53 | 54.17 | 25.01 | 55.67 | 18.06 | 51.67 | 19.30 | 46.08 |
| + Retrieval w/ Ref. | OpenAI Ada | 27.95 | 56.59 | 28.33 | 58.39 | 21.57 | 55.07 | 24.73 | 51.51 |
| StarCoder-15.5B | - | 15.72 | 48.16 | 18.28 | 53.23 | 11.86 | 43.53 | 8.54 | 34.33 |
| + Retrieval | BM25 | 24.77 | 55.57 | 25.95 | 57.74 | 14.09 | 39.50 | 18.04 | 44.38 |
| + Retrieval w/ Ref. | BM25 | 30.24 | 59.46 | 29.73 | 60.47 | 17.55 | 42.18 | 24.38 | 49.09 |
| + Retrieval | UnixCoder | 20.41 | 46.66 | 19.26 | 47.30 | 15.55 | 45.91 | 15.61 | 38.86 |
| + Retrieval w/ Ref. | UnixCoder | 25.25 | 55.42 | 25.90 | 57.88 | 13.32 | 39.13 | 16.23 | 42.36 |
| + Retrieval | OpenAI Ada | 25.55 | 56.23 | 26.65 | 57.68 | 14.15 | 39.99 | 19.98 | 46.45 |
| + Retrieval w/ Ref. | OpenAI Ada | 29.64 | 58.64 | 30.86 | 60.73 | 18.12 | 42.77 | 27.96 | 53.53 |

Table 9: Identifier Match evaluation results of various methods in retrieving cross-file context (cf. Table 4).

Different from Figure 9, Figure 10 no longer retrieves the similar usage of predicting APIs, and rather, it retrieves the implementation details of the cross-file dependencies. Concretely, the cross-file context collects the member function, `store_by_text`, of class `EntitySessionStorage`, which is imported by the current completing file and instantiated as `self.entity_repository`. Without such cross-file context, the model hallucinates a wrong function usage that is not defined in class `EntitySessionStorage`. In contrast, when the cross-file is prepended to the prompt, the model successfully predicts the ground truth, empirically revealing the limitation of existing datasets by only feeding the current-file context, which will consequently underestimate code LMs' capacity.

# E   Limitations

**Zero-shot Evaluation**   Our benchmarking was done in a zero-shot fashion. We didn't perform the few-shot study as the max sequence length of most benchmarked models is quite limited for prepending additional examples to the prompt. Thus, the performance will be limited as the format of cross-file context is never seen by the model during both training and prompting. We hope that CROSSCODEEVAL encourages future research to investigate methods for efficiently retrieving and incorporating cross-file context into the model.

**Cross-file Context Retrieval Quality**   Prepending cross-file context to the prompt has shown significant improvement to the code LM's performance. However, as we have analyzed and identified in Section 3.5, the RG retrieval framework is not perfect. Due to its fixed length of context window and token-based similarity calculation, RG sometimes retrieves useless information and fails to help code LMs for better generation. As for future work, we are expecting a more advanced retrieval approach to replace RG for more accurate cross-file context.

**Memorization**   Code LMs were trained on a vast amount of unlabeled code. There is no way we could ensure that all models didn't see the evaluation data in the past. We take our best effort by excluding popular packages from annotation (see Section 2.1). Despite that, we suggest researchers and practitioners be cautious in interpreting the results with potential memorization in mind. Future research in incorporating cross-file context may also consider deduplicating the training data with CROSSCODEEVAL, e.g., through methods in Lee et al. (2022).

## Cross-file Context

```
# ...
# the below code fragment can be found in:
# examples/constraints.py
#     def __init__(self, prompt, can_follow):
#         super().__init__()
#         self.context = self.new_context(prompt)
#         self.can_follow = can_follow
#     def step(self):
#         # Generate proposed token.
#         token = self.sample(llp.Transformer(self.context),
#                             proposal=self.locally_optimal_proposal())
# ...
```

## In-file Context

```python
import llamppl as llp
import numpy as np

class Infilling(llp.Model):
    def __init__(self, words):
        super().__init__()
        self.s = words.pop(0)
        self.ctx = self.new_context(self.s)
        self.remaining_segments = [self.llama.tokenize(w) for w in words]
    def start(self):
        self.step()

    def step(self):
        # Generate a token
        n = self.sample(llp.Geometric(0.5)) + 1
        for _ in range(n):
            self.s += self.sample(llp.[CURSOR_POSITION]
```

| ❌ **w/o** Cross-file Context | ✅ **with** Cross-file Context |
|---|---|
| `Categorical(self.ctx.get_token_probs())` | `Transformer(self.ctx)` |

Figure 9: Qualitative Example 1: the cross-file context provides a similar usage of the predicting API.

**Cross-file Context**

```
# ...
# the below code fragment can be found in:
# doccano_mini/storages/entity.py
# ...
#         entities = self.storage.get_state("entities")
#         return entities.get(text, [])
#     def store_by_text(self, text: str, entities: List[Entity]) -> None:
#         current_entities = self.storage.get_state("entities")
#         current_entities[text] = entities
#         self.storage.set_state("entities", current_entities)
# ...
```

**In-file Context**

```python
from typing import Dict, List
...
from doccano_mini.storages.entity import EntitySessionStorage
from doccano_mini.storages.stepper import StepperSessionStorage

class NamedEntityRecognitionPage(BasePage):
    example_path = "named_entity_recognition.json"

    def __init__(self, title: str) -> None:
        super().__init__(title)
        self.types: List[str] = []
        self.entity_repository = EntitySessionStorage()
        self.stepper_repository = StepperSessionStorage()

    def define_entity_types(self):
        ...

    def annotate(self, examples: List[Dict]) -> List[Dict]:
        if len(examples) == 0:
            return []
        types = self.define_entity_types()
        selected_type = st.selectbox("Select an entity type", types)
        step = self.stepper_repository.get_step()
        text = examples[step]["text"]
        entities = self.entity_repository.find_by_text(text)
        entities = st_ner_annotate(selected_type, text, entities, key=text)
        self.entity_repository.[CURSOR_POSITION]
```

❌ **w/o** Cross-file Context          ✅ **with** Cross-file Context

```
save(entities)                  store_by_text(text, entities)
```

Figure 10: Qualitative Example 2: the cross-file context provides the definition of the predicting API.

