# OpenReview forum: "CrossCodeEval: A Diverse and Multilingual Benchmark for Cross-File Code Completion"
_NeurIPS.cc/2023/Track/Datasets_and_Benchmarks — NeurIPS 2023 Datasets and Benchmarks Poster_

### Official Review · Reviewer_BzTx · 2023-07-10
**Novel Cross-File code completion dataset, but limited to line completion and without execution-based metrics**

**Rating:** 6
**Confidence:** 4
**Clarity:** Overall, the paper is well written an…

**Strengths:**

1. Novel dataset: The paper's introduction of the CROSSCODEEVAL dataset fills a gap in the literature by providing a benchmark that goes beyond single-file code completion tasks. By focusing on real-world software development scenarios involving multiple files and cross-file dependencies, the dataset offers a more realistic evaluation environment.

2. Evaluation of existing models: The paper conducts experiments using popular code language models like CodeGen and StarCoder. By evaluating these models on the CROSSCODEEVAL dataset with different levels of cross-file context, including no context, reference-guided context, and oracle reference-guided context, the paper provides valuable insights into the performance of state-of-the-art models in handling code completion tasks that require cross-file understanding.

3. Potential for evaluating retrieval strategies: CROSSCODEEVAL dataset can be used to evaluate novel retrieval strategies aimed at improving code completion. Retrieval strategies can enhance the accuracy and efficiency of code completion by leveraging relevant code snippets or documents from the repository. By considering the cross-file context and exploring retrieval-based approaches, the dataset opens up possibilities for further research and development in code completion techniques.


**Additional Feedback:**

None

**Correctness:**

The paper provides a detailed description of the static analysis-based approach used to create the CROSSCODEEVAL dataset. The authors explain their method for identifying code fragments that require cross-file context by replacing import statements with empty classes and detecting undefined names through static analysis. This approach is well-defined and seems sound from a technical standpoint.

Additionally, the authors mention that they perform several filtering steps and manual analyses to ensure the quality and correctness of the dataset.


**Documentation:**

The documentation provided in the Supplemental Material pdf.

**Ethics:**

Ethics concerns are discussed in Sec. 5.6 of the supplemental material.

**Limitations:**

1. Limited to Line/Statement completion: One weakness of the paper is that the CROSSCODEEVAL dataset only focuses on line/statement completion. This narrow focus on completing individual lines limits the dataset's potential applications and does not capture more complex code generation tasks, such as generating entire methods or functions. Expanding the dataset to include tasks beyond line completion would make it more versatile and applicable to a wider range of code completion scenarios.

2. Lack of evaluation with newer models: Another weakness is that the evaluation section does not include newer models such as OpenAI GPT-3.5. As the field of code completion evolves rapidly, it would be beneficial to include evaluations with newer models to provide a more comprehensive assessment of the state-of-the-art performance on the CROSSCODEEVAL dataset.

3. Lack of execution-based metrics: The paper mentions that the evaluation metrics used are Exact Match and Edit similarity. While these metrics can provide some insights into the correctness and similarity of generated code, they do not capture the execution-based aspects of code completion. Execution-based metrics, such as test execution or runtime behavior analysis, can provide a more comprehensive evaluation of code completion models by considering the functional correctness and performance of the generated code. Incorporating execution-based metrics into the evaluation would strengthen the assessment of code completion models on the dataset.


**Opportunities For Improvement:**

Opportunities for improvement in the paper can be identified in the weaknesses section. The most important would be to expand the dataset to allow more challenging code generation tasks such as entire method completion (which would require cross-file context and understanding).

**Relation To Prior Work:**

Related works are discussed and the paper differs from previous contributions.

**Summary And Contributions:**

The paper introduces CROSSCODEEVAL, a benchmark dataset for code completion that requires cross-file contextual understanding. Existing evaluation datasets focus on code completion within a single file, which does not accurately reflect real-world software development scenarios where repositories consist of multiple files with cross-file dependencies. CROSSCODEEVAL addresses this limitation by using real-world, open-source repositories in Python and Java to create examples that strictly require cross-file context to complete the code accurately. To identify code fragments that require cross-file context, the paper proposes a static-analysis-based approach. The paper focuses on Python and Java as the target programming languages, using Pylint and the javac compiler for static analysis, respectively. The resulting CROSSCODEEVAL dataset is diverse and multilingual, consisting of code completion examples in Python and Java. The dataset includes code prompts with cursor positions and references that extend from the cursor position to the end of the statement. The examples have the key property that the statement to be completed must include at least one use of a local API defined in the software repository. Experimental results using state-of-the-art code language models, such as CodeGen and StarCoder, demonstrate that CROSSCODEEVAL poses a challenge when cross-file context is not provided to the models. Performance significantly improves when the cross-file context is included in the prompt. The paper concludes by highlighting the potential of CROSSCODEEVAL to promote research and development of code completion models that better reflect real-world software development scenarios, where cross-file context is crucial.

---

> ### Author Response · Authors · 2023-08-23
> **Response to BzTx**
>
> Thank you for the thoughtful comments and great questions!
>
> - __Line/Statement Completion:__ Please see “G3: Code Completion Scope” in the general response.
>
> - __Latest code LMs:__ Following your suggestion, we have evaluated GPT3.5 on CrossCodeEval. Please see “G2: G2: Latest SOTA Code LM Evaluation” in the general response. In addition to GPT 3.5, we also evaluated CrossCodeEval on two very recent open-source SOTA code LMs: StarCoder (released 2023/05) and CodeGen2.5 (released 2023/07). Both models report promising performance in code completion. We hope this will provide a comprehensive picture in evaluation.
>
> - __Execution metrics:__ Existing benchmarks, such as HumanEval and MBPP, include execution-based metrics, but they (1) focus mostly on programming and algorithm questions, and (2) only include isolated functions (i.e., no inter-procedural dependencies) as samples, where function-level unit tests are easy to collect and execute. In contrast, CrossCodeEval collects samples from real-world software projects in different programming languages, where execution configurations are much varied across projects and languages, so implementing a unified execution framework for all CrossCodeEval samples remains a challenge. Thus, we regard it as future work.

---

### Official Review · Reviewer_1pD2 · 2023-07-13
**CROSSCODEEVAL: A Diverse and Multilingual Benchmark for Cross-File Code Completion**

**Rating:** 7
**Confidence:** 3

**Strengths:**

1. Cross File Scope: CROSSCODEEVAL's strengths lie in its cross-file scope, allowing for comprehensive evaluation of code language models in Python and Java.


2. Automated Data Generation based on Static Analysis: The methodologies automate dataset generation through static analysis, reducing manual effort and ensuring a large and representative dataset.


3. Caution for Data Leakage: They address data leakage concerns by focusing on recently created, non-fork repositories, avoiding overlap with existing models' training data.


4. Quality Control Process: A quality control process is implemented, verifying absence of dataset overlap and discarding false positives through static analysis.


**Additional Feedback:**

It is better to add the examples to test whether the model can do well under the reversed setting that the function call is given and predict the function as a callee across the file context. And the name of the benchmark is confusing because in general the term ‘multi-lingual’ is only used if the number of languages is larger than or equal to 3. Thus, if you want to use the term ‘multi-lingual’, you should mention the plan of extension in terms of programming languages.

**Clarity:**

The necessity of the suggested benchmark is well explained and the details of the data construction process are enough to understand why and how they do it. However, the design of the experiment is not explicit. Besides since Table 2 for each CodeLLM is not very readable, they need to show only the core parts of that table.


**Correctness:**

which is widely used for natural language processing tasks. However, the current SoTA for the code completion task is based on Codex. Therefore, the authors need to experiment with Codex(code-cushman-002) also. The experiments are conducted in the zero-shot setting, aligning with standard practices for evaluating pre-trained language models. This zero-shot setting is especially suitable for this task because if you use other examples under a few-shot setting, the context may become really long.

Evaluation Metrics: The benchmark utilizes two main categories of evaluation metrics: 1) code match and 2) identifier match. Unlike the code completion task, this CROSSCODEEVAL's core part is to use the already defined code. So these metrics provide an appropriate assessment of the models' performance in terms of generating accurate code and predicting correct APIs in code snippets.

Greedy Search: Current trend for design of code completion is prediction and reranking. So it is not enough to say that current LLM does not do well on cross-file code completion under the greedy search setting. Therefore, authors need to experiment under the setting of prediction and reranking settings.


**Documentation:**

Overall, the authors give enough information about the dataset. They show the details about the process of data construction and their considerations. In particular, they show the splitting point between prompt and reference using static analysis with figures. However, it is difficult to understand how the backward mapping operates. So it is necessary to clarify the mechanism of this.

And according to figure 3, there is no instruction for the code completion. It is necessary to explain the reason or experiment results since the code model may be confused if there is no instruction and it is the general setting.

**Limitations:**

Since the majority of ground truth consists of a single statement, it only evaluates the cross file context setting. It cannot test the competence of the model to complete the given code enough. Thus they should mention about this situation and, for the accurate evaluation, the evaluator should use the existing code completion benchmark together

**Opportunities For Improvement:**

1. Limited Language Support: While CROSSCODEEVAL is a multilingual benchmark, it currently only supports Python and Java. This limitation restricts the evaluation and applicability of the benchmark to these two programming languages. It would be beneficial to expand the benchmark to include other popular programming languages to provide a more comprehensive evaluation.

2. Focus on Scope Completion: The current version of CROSSCODEEVAL focuses on scope completion tasks, where the completion targets are statements involving local APIs. While this is an essential aspect of code completion, it does not cover other aspects such as function completion tasks. Expanding the benchmark to include additional completion tasks would provide a more comprehensive evaluation of code language models.

3. Unidirectional Setting: Based on the examples they show, there is no reversed setting that makes LLM implement code from the call of that code. In the real scenario, it is common to implement new functions while coding others. So it is necessary to add those kinds of problems and if they are already included, then authors should specify it in the paper.


**Relation To Prior Work:**

The authors emphasize that the current benchmarks only evaluate the ability of Code Language Model only in terms of single file context but their proposal can cover the cross-file context for the code completion in Abstract and Introduction Sections. They also specify many code completion benchmarks in Related Works and try to compare some of these with their suggestion. However, for the others, they just enumerate the benchmarks and do not compare them with theirs. Thus, it would be better if the  authors could discuss a comparison with other benchmarks.

**Summary And Contributions:**

This paper highlights the limitations of existing code completion evaluation settings and proposes a new benchmark called CROSSCODEEVAL to address these limitations. The new benchmark focuses on evaluating code language models' ability to utilize cross-file context for code completion in realistic software development scenarios. The dataset comprises samples from Python and Java repositories, carefully curated and filtered to minimize overlap with existing training data, ensuring unbiased evaluation. The evaluation includes popular code language models such as CodeGen, CodeGen2, SantaCoder, and StarCoder, varying in parameter sizes. Results show that these models perform poorly when relying solely on current-file context but show significant improvement when cross-file context is included. CROSSCODEEVAL is expected to serve as a valuable resource for evaluating code language models' utilization of cross-file context and facilitate future research in this area. The dataset will be made publicly available upon acceptance.

---

> ### Author Response · Authors · 2023-08-23
> **Response to 1pD2**
>
> Thank you very much for your support and the detailed suggestions!
>
> - __Language Extension:__ We fully agree! Please see “G1: Incorporating More Programming Languages” in the general response.
>
> - __Completion Scope:__ Thanks for raising it. Please see “G3: Code Completion Scope” in the general response.
>
> - __Simultaneous Editing:__ CrossCodeEval focuses on highlighting the importance of cross-file context for code completion which most existing benchmarks overlooked. Therefore, as a proof-of-concept, CrossCodeEval formulates the problem by imitating one certain developing scenario that requires cross-file context, where we assume that the callee is finalized and the user is completing the caller only. We agree with the reviewer that another interesting scenario where the user edit caller and callee interactively is also possible, which is a more challenging case, and we regard it as future work.
>
> - __Evaluating SOTA Models:__ As CodeX is no longer accessible (See “In March 2023, OpenAI shut down access to Codex…” in: https://en.wikipedia.org/wiki/OpenAI_Codex), we follow OpenAI’s migration guide (https://platform.openai.com/docs/deprecations/) and evaluate GPT3.5-Turbo on CrossCodeEval. Please see “G2: Latest SOTA Code LM Evaluation” in the general response for details. In addition to GPT 3.5, we also evaluated CrossCodeEval on two very recent open-source SOTA code LMs: StarCoder (released 2023/05) and CodeGen2.5 (released 2023/07). Both models report promising performance in code completion. We hope this will provide a comprehensive picture in evaluation.
>
> - __Greedy Search:__ Please see “G4: Greedy vs Sampling + Reranking” in the general response.

---

### Official Review · Reviewer_C4R7 · 2023-07-20
**Should we instead benchmark lots of different code-retrival approaches with 1-2 fixed LMs?**

**Rating:** 6
**Confidence:** 5
**Correctness:** No correctness issues.

**Strengths:**

An important problem. We definitely need code generation benchmarks that are
longer than one function.


**Additional Feedback:**

None

**Clarity:**

The paper is mostly clear. I had a somewhat hard time understanding L74--L81,
which is the heart of the dataset construction approach.


**Documentation:**

Yes.

**Ethics:**

No ethical concerns.

**Limitations:**

Limitations and potential impacts are appropriately addressed.


**Opportunities For Improvement:**

- I don't see how the Oracle approach makes sense. If I understand correctly,
  it is *almost* turning code completion into a multiple-choice (5 choices)
  problem for the LM. Why is this reasonable?

- Why greedy decoding, when virtually all LM evaluations use sampling?

- It would be nice to see more languages than just Python and Java represented in a
  multi-language benchmark.

**Relation To Prior Work:**

Yes.

**Summary And Contributions:**


This submission presents a benchmark for cross-file code completion. Care is
taken to ensure that the benchmark problems do not overlap with The Stack,
which is a widely used training set for Code LLMs.

The *cross-file code completion* task is the task of generating code
completions for program text, where the program spans several files.
The hard case, which this submission benchmarks, is when the masked line
depends on the contents of other files. Thus, without cross-file context,
models perform poorly, which the paper demonstrates with several models.

My main concern is that the benchmark seems to have fixed cross-file contexts
and varies the LLMs. However, contemporary research is working toward picking
the right context to retrieve. So, it may be benchmarking the wrong piece of
the problem. Maybe we should be benchmarking several different approaches to
retrieval with 1-2 fixed models instead.

---

> ### Author Response · Authors · 2023-08-23
> **Response to C4R7**
>
> Thanks for your thoughtful feedback and questions!
>
> - __Oracle RG:__ We apologize for the confusion. In general, the oracle RG is proposed to quantify and analyze the upper bound impacts of cross-file context retrieved by the RG framework.To make the approach more understandable, we make three revisions: (1) We update Figure 3 with more details regarding the oracle context and the formats of different prompts (2) We significantly revise the text in Section 3.3 according to the updated figure for better illustration. (3) We rename the confusing abbreviation RG and Oracle RG as “Cross-file Context” and “Oracle Context” to make them more intuitive, and update the term across the paper. Please read the revised Section 3.3 in the manuscript and let us know if you need more details.
>
> - __Greedy Decoding:__ Please see “G4: Greedy vs Sampling + Reranking” in the general response.
>
> - __More Programming Languages:__ We fully agree! Please see “G1: Incorporating More Programming Languages” in the general response.
>
> - __Clarity Improvement:__ Apologize for the confusion. Following your suggestion, we add more clarifications to Section 2.2 (previously L74-81 as mentioned). Please check the revised manuscript and let us know if you have further questions.

---

### Official Review · Reviewer_t83P · 2023-07-20
**benchmark is useful but presentation needs improvement**

**Rating:** 6
**Confidence:** 5
**Correctness:** I don't have any correctness concerns.
**Clarity:** The presentation should be improved i…

**Strengths:**

This paper contributes a benchmark that fills an evaluation need in the code completion community. It is difficult to assess how well models perform on code completion that requires knowledge of other files within a repository. A strength of the proposed benchmark is the care taken to ensure that it does not overlap with the Stack. I also thought that the authors showed careful attention to detail in how they constructed the dataset, specifically, in ensuring that the model can't infer the call from prior context, and in the careful human annotation effort for additional verification.

**Additional Feedback:**

See above.

**Documentation:**

Yes, the dataset is well-documented.

**Ethics:**

No, I do not have any ethical concerns.

**Limitations:**

The authors do not have a section discussing the limitations of their work. This is another aspect of presentation that could be improved.

**Opportunities For Improvement:**

Although I think the benchmark is a good contribution and was carefully constructed, I had a number of concerns related to its presentation:

+ Most seriously, the discussion of the RG and Oracle RG evaluation methods was extremely difficult to follow. I am not sure I fully understand the Oracle RG method. Showing examples of all three prompting formats in the main body of the paper would be helpful; additional details in the supplemental materials would also be welcome.
+ The review of previous code benchmarks is fairly comprehensive, but it is presented as a list, without substantive discussion of how this work improves on prior work. This is particularly important with respect to previous repository-level benchmarks.
+ The authors cite MBXP and CodeGeeX as multilingual extensions of MBPP and HumanEval respectively; however, Cassano et al. 2023 extends both benchmarks to HumanEval and MBPP, and appeared before MBXP.
+ It would be better to mention Python and Java explicitly rather than referring to the benchmark as multilingual; multilingual implies more than two languages.
+ The text in the caption for Figure 1 was difficult to follow.



**Relation To Prior Work:**

As discussed above, the discussion of previous repository-level benchmarks should be revised to explicitly address how this work differs from previous work.

**Summary And Contributions:**

This paper proposes a repository-level benchmark for code completion in Java and Python. They construct a dataset for each language of code snippets with cross-file calls (references to methods or classes in other files in the same repository). They benchmark state-of-the-art code completion models on this task, and find that it is a challenging benchmark.

---

> ### Author Response · Authors · 2023-08-23
> **Response to t83P**
>
> Thanks for your meticulous review and feedback!
>
> - __RG & Oracle RG:__ We apologize for the confusion. To make the approach more understandable, we make three updates: (1) We update Figure 3 with more details regarding the oracle context and the formats of different prompts (2) We significantly revised the text in Section 3.3 according to the updated figure for better illustration. (3) We rename the confusing abbreviation RG and Oracle RG as “Cross-file Context” and “Oracle Context” to make them more intuitive, and update the term across the paper.  Please read the revised Section 3.3 in the manuscript and let us know if you need more details.
>
> - __Discussion of Existing Benchmarks:__ We expand and re-organize Section 4.1 to discuss the differences with existing datasets. Please check the revised Section 4.1 in the updated manuscript, and we have added the missing MultiPL-E citation — thanks for catching it!
>
> - __Language Extension:__ We fully agree! Please see “G1: Incorporating More Programming Languages” in the general response.
>
> - __Figure 1 Caption:__ Apologize for the confusion. In the revised manuscript, we update the caption with more clear reference to different parts of the figure. Some clarification here: in-file context indicates code snippets in the current file before the cursor position, shown in the left part of the figure. Cross-file context indicates the context from other files, e.g., the the right part of the figure. The format of prompts with cross-file context is illustrated in the updated Figure 3. Please let us know if this is clear!
>
> - __Limitation Section:__ Thanks for bringing it up! We add a limitation section in the Supplementary Materials to discuss the limitation of CrossCodeEval. Please check Section 5 of the revised Supplementary Materials.

---

> > ### Comment · Reviewer_t83P · 2023-08-24
> >
> > Thank you for your response and the revisions! The updated Figure 3 is very helpful, as is the extra text in the Section 3.3.

---

### Official Review · Reviewer_K8CN · 2023-07-28
**Present a code completion benchmark with long-term dependencies detected. Further studies needed.**

**Rating:** 6
**Confidence:** 4
**Clarity:** Yes.

**Strengths:**

The strengths of this paper lie in its meticulous approach to data collection and post-processing. Preliminary experiments have validated the effectiveness of the benchmark in measuring long-distance dependencies. As the field progresses, we anticipate an increase in the number of benchmarks in this area, which will be instrumental in assessing the actual performance of code completion models.

**Additional Feedback:**

No.

**Correctness:**

The evaluation method employed is fundamentally sound, but it would be beneficial to incorporate an ablation study. This would ensure that the model's improvement stems from long-term dependencies rather than from the inclusion of additional code information.

**Documentation:**

No.

**Ethics:**

It is necessary to ensure that the sensitive information of the dataset is deleted.

**Limitations:**

Yes.

**Opportunities For Improvement:**

Firstly, in terms of data collection, long-term dependencies can be demonstrated through the invocation of custom methods and other aspects. For instance, a current method and its logic may closely resemble the logic of a previous method, requiring only simple modifications, thereby exhibiting long-term dependencies.

Secondly, concerning experiments, this study could conduct additional tests to eliminate interference items. For example, using random code snippets as a pseudo-long-term context could serve as a baseline, ruling out the possibility that random code chunks might provide substantial information to the model.

**Relation To Prior Work:**

Yes.

**Summary And Contributions:**

This paper observes that benchmarks in the current coding field seldom consider long-term dependencies, despite their critical importance for code completion in extensive code projects. Consequently, we have developed a high-quality benchmark for testing large code models, incorporating data collection, filtering, and extraction of long-term dependent contexts. Through a series of experiments, we have demonstrated that models such as CodeGen2 and StarCode significantly improve the benchmark when provided with long-term dependent contexts. This validates the benchmark's ability to measure the quality of long-term dependent contexts accurately and underscores the necessity for large Code Language Models (LLM) to manage long-term dependencies.

---

> ### Author Response · Authors · 2023-08-23
> **Response to K8CN**
>
> Thank you for the insightful comments and questions!
>
> - __Different Perspectives of Cross-file Context:__ This is a great suggestion to identify long-dependeicies based on logical similarity across code fragments. However, it is difficult to define logical similarity across code fragments, and thus we chose to focus on identifying dependencies via custom method invocations. We will explore opportunities to benefit LMs by providing logically similar code fragments within the repository as our future work.
>
>
> - __Pseudo Long-term Context as Baseline:__ We add the experiments studying the effects of random code snippets as “pseudo-long-term” context and verify they are not helpful, since there are typically thousands of cross-file chunks, and randomly picking a few code chunks could not provide useful references. Please see the revised Supplementary Material Section 4.3 for details.

---

### Author Response · Authors · 2023-08-23
**General Response to Reviewers**

We thank all the reviewers for their insightful and constructive suggestions!

__Summary__

We have revised the paper based on the suggestions, and we hope we have addressed concerns from all reviewers. Please check the revised manuscript and supplementary materials (changes are marked in red).

Some highlights include:
- Expanding CrossCodeEval to 4 languages: Python, Java, TypeScript and C\#.
- A comprehensive evaluation on the latest public and proprietary SOTA models e.g. StarCoder (released 2023/05), CodeGen 2.5 (released 2023/07) and GPT 3.5.
- More polishment and ablation studies, e.g., on sampling and pseudo-context.

We address common questions as the general response below, and respond to the rest individually.

__G1: Incorporating More Programming Languages (t83P, C4R7, 1pD2)__

We have extended CrossCodeEval to two additional widely-used languages: TypeScript and C#. We follow similar method in Python and Java in dataset curation. We obtained 7868 and 1300 prompts respectively for TypeScript and C# and benchmarked their performance with various models. We observed similar trends in performance as in Python and Java. Please read the revised Section 2 in the manuscript for details. Meanwhile, please let us know if there are any additional languages of interest, and we can look into it to extend CrossCodeEval further.


__G2: Latest SOTA Code LM Evaluation (1pD2, BzTx)__

We evaluate OpenAI models on CrossCodeEval Python and Java using GPT3.5-Turbo. Overall, we see a similar trend compared with public models, and the performance is on par with best public models. In addition, we also added CodeGen 2.5 to the evaluation. CodeGen 2.5 is high performing and was released only a month ago (2023/07). In all, we hope our evaluation now is comprehensive enough to demonstrate the features of our benchmark. Please read the revised Section 3.4 for details.

__G3: Code Completion Scope (1pD2, BzTx)__

We discuss why we considered statement-level completion in CrossCodeEval and didn’t focus on a larger scope, e.g., function-level completion.

First, developers prefer single-line code completions than multi-line suggestions due to their effectiveness and alignment with human intentions [1]. Existing models excel at predicting statements, aiding programmers in speeding up coding tasks. Lengthy suggestions disrupt the developers’ logic and work flow and might not align with intent [1].

Second, statement-level code completion helps assessing code LMs’ capability on completing statements involving local API/class usage in a more controlled and isolated way. As we rely on metrics such as exact match or edit similarity, moving from statement-level to function-level also makes the evaluation less insightful as function-level scope could be very long and consists of complex function implementation.

Third, inclusion of function completion tasks in CrossCodeEval require us to support evaluation of functional correctness (through unit testing), which is very challenging given we create examples from hundreds of repositories (to ensure diversity and freshness) and cover multiple languages. However, we believe, with the release of CrossCodeEval, there will be effort to enable functional accuracy evaluation using a fraction of the repositories if not all of them can be used (by ensuring coverage of the targeted functions).

[1] Shraddha Barke, Michael B. James, and Nadia Polikarpova. 2023. Grounded Copilot: How Programmers Interact with Code-Generating Models. OOPSLA ’23. https://doi.org/10.1145/3586030

__G4: Greedy Seach vs Sampling + Reranking (C4R7, 1pD2)__

We included results using nucleus sampling with mean-log-p reranking, a commonly used setting for LLMs. We noticed that (1) the performance difference between greedy decoding and sampling with re-ranking is marginal, and (2) the cross-file context is helpful regardless of the decoding algorithms. Please see the revised Supplementary Materials Section 4.2 for details.

---

### Decision · Program_Chairs · 2023-09-22

**Decision:**

Accept (Poster)

**Comment:**

The paper first analyzes the pitfalls of previous code completion evaluation setting (single file) and proposes CROSSCODEEVAL a benchmark dataset for code completion that requires cross-file contextual understanding. This dataset uses real-world, open-source repositories in Python and Java to create examples that strictly require cross-file context to complete the code accurately. To identify code fragments that require cross-file context, the paper proposes a static-analysis-based approach. In the revised version, it covers Python, Java, typescript and C# programming languages with well curated samples to ensure an unbiased, an proper training set. The evaluation covers CodeGen, CodeGen2, SantaCoder, and StarCoder code language models in different size.  Experimental results demonstrate that CROSSCODEEVAL poses a challenge when cross-file context is not provided to the state-of-the-art code language models. Performance significantly improves when the cross-file context is included in the prompt.

All reviewers agree on the acceptance of the paper, and highlight the importance of the problem and the new research opportunities such a benchmark will open.
In addition, the revised version of the authors tackles the minor criticisms and suggestions such as recent LLM and more programming languages.